# Beef Toughness and the Amount of Greenhouse Gas Emissions as a Function of Localized Electrical Stimulation

**DOI:** 10.3390/foods13010037

**Published:** 2023-12-21

**Authors:** Dawoon Jeong, Young Soon Kim, Hong-Gun Kim, Inho Hwang

**Affiliations:** 1Department of Animal Science, Jeonbuk National University, Jeonju 54896, Republic of Korea; ddown98@jbnu.ac.kr; 2Institute of Carbon Technology, Jeonju University, Jeonju 55069, Republic of Korea; kyscjb@jj.ac.kr (Y.S.K.); hkim@jj.ac.kr (H.-G.K.)

**Keywords:** localized electrical stimulation, Hanwoo beef, aging, tenderness, CO_2_ emission, Warner–Bratzler shear force

## Abstract

This study investigated the effect of localized electrical stimulation on Hanwoo beef quality. It focused on the chemical and physical properties of the Longissimus thoracis (LT) and Biceps femoris (BF) muscles, and it explored the implications of carbon dioxide (CO_2_) reduction achieved by accelerating tenderization via localized electrical stimulation. The results show that the application of localized electrical stimulation (45 V) had no significant impact on the TBARS (thiobarbituric acid reactive substances) of either the LT muscle or the BF muscle. Localized electrical stimulation and aging treatments had a significant effect on meat tenderness in the LT and BF muscles, but there was no interactive effect. In particular, the WBsf (Warnar–Bratzler shear force) at 2 days of aging of the electrically stimulated BF muscle was 5.35 kg, which was lower than that of the control group (5.58 kg) after 14 days of aging; however, the effect of WBsf reduction due to aging in the LT muscle was higher than the localized electrical stimulation effect. Estimating CO_2_ mitigation from a shorter feeding period for Hanwoo steers from 31 months to 26 months may reduce 1.04 kg of CO_2_-eq emissions associated with the production of a single kilogram of trimmed beef. In conclusion, localized electrical stimulation improved the tenderness of Hanwoo beef and reduced CO_2_ emissions.

## 1. Introduction

With global warming accelerating and abnormal climate phenomena occurring worldwide, concerns about excessive greenhouse gases (GHG) are mounting [1]. The FAO’s “Tackling climate change through livestock” reported that livestock has a huge impact on global warming [2]. In response, sustainable ethical consumption among consumers is fashionable, and they have recognized the need to improve the environmental sustainability of beef production [3]. To date, various methods designed to reduce the environmental impact of livestock production have been proposed. Capper suggested the importance of improving cattle productivity through genetic improvements, feed, and growth-enhancing technologies [4]. Beyond the farm gate, GHG generated by the refrigerant gas used for energy users and refrigeration/frozen storage/distribution account for most of all GHG [5]. It has been reported that GHGs generated from refrigerated/frozen distribution take up the largest portion after the farm gate stage because a constant low temperature must be maintained until the meat is consumed [6]. Many researchers have suggested technologies such as feed additives that would reduce methane gas emissions from livestock, techniques for reducing nitrogen discharge, and technology for precise feeding [7]. However, all of these technologies are applied at the farm stage of the beef production chain.

Korean consumers prefer highly marbled and tender meat [8], so the Korean beef grading system focuses on long-term fattening to produce premium beef with a high marbling score [9]. For Hanwoo cattle, long-term fattening typically requires over 30 months before shipment, as after 26 months of age, the intramuscular fat increases but there is a decrease in daily weight gain and meat yield [10]. The longer it takes to rear the cattle, the higher the GHG emissions per functional unit [11]. However, because domestic consumers still desire this tender meat, there is a big gap between the environmental burden and the demand for highly marbled beef.

Electrical stimulation and aging techniques are among the most commonly used means of improving meat tenderness [12]. Electrical stimulation has been in use since the 1950s as a means of accelerating post-mortem glycogen breakdown, improving meat tenderness, and enhancing meat color [13]. It is still utilized in diverse species in the United States, Australia, New Zealand, and Europe, and its value has been confirmed by many researchers [14]. Improving the tenderness through electrical stimulation not only physically improves the meat but also increases the activity of muscle protein-degrading enzymes, which may lead to faster aging [15] and could increase the number of components related to the taste of the meat [16]. According to a study, when 24 retail cuts of Hanwoo beef were cold-aged, aging continued even after 10 days across all parts of the animal, and the initial shear force and tenderizing speed varied depending on the muscle. Moreover, it took six days for the tenderloin to reach a shear force of 3.2 kg (the level consumers perceive as tender) and twenty-eight days or more for the outside round [17]. Previous studies reported that aging beef and pork increases taste intensity, that peptides of less than 17 kDa are more present in aged meat than in unaged meat [18], and that aging improves tenderness, flavor, taste satisfaction, and overall palatability [19].

Electrical stimulation technology that can be applied to a carcass is currently being industrially utilized in many countries [14]. Commonly, electrical stimulation is performed on the whole carcass at the slaughter stage [20]. However, no study has been conducted on the environmental impact of beef production with this tenderization technology. In addition, almost all research on beef production and consumption has focused exclusively on quality improvement [4,5,6,7], with research on CO_2_ emissions from production and/or consumption being neglected. In this study, we investigated the possibility of shortening the fattening period for cattle and the aging period for beef through easily applicable localized electrical stimulation and estimated the relationship between the meat tenderness improvement technology and CO_2_ emissions from the beef.

## 2. Materials and Methods

### 2.1. Animals and Treatment

Twenty Hanwoo steers (average age of 31 months) were used in this experiment, comprising three heads of grade 2 quality, seven heads of grade 1 quality, seven heads of grade 1+ quality, and three heads of grade 1++ quality. All animals were shipped from feedlots to a commercial slaughterhouse and held in lairage following normal plant operating guidelines and Korea Animal Plant Quarantine Agency of Ministry of Agriculture, Food and Rural Affairs regulations for beef slaughter. These animals were slaughtered following the regulation for beef slaughter, and then immediately moved to a chilling room and stored at 4 °C. In the chilling room, the Longissimus thoracis (LT) and Biceps femoris (BF) muscles from the right side of the carcasses were electrically stimulated using an electrical stimulation device with a specialized probe. Figure 1 shows the design of the localized electrical stimulation probe and the locations where it was inserted into the carcasses. The left side of the carcasses of the same animals were used as the control. Electrical stimulation was performed with alternative current (AC), 45 V for 1 min. All electrical stimulation was completed within 30 min of slaughter. After 24 h in the chilling room, all carcasses were graded according to the Korean Beef Carcass Grading System [21]. The LT and BF muscles were taken from both the left and right sides of the carcasses at 36 h postmortem, and then transported to the Jeonbuk National University Meat Science Laboratory in a chilled condition. These muscles were divided and vacuum-packed using a household vacuum-packaging machine (VS2011, Food saver^®^, Atlanta, GA, USA), and then stored for 2 and 14 days at 4 °C until analyzed. The packaging material used for vacuum packaging was a 4-layer (LLDPE/LDPE/LDPE/Nylon) film-material vacuum bag.

### 2.2. Aging Speeder through Localized Electrical Stimulation

An electrical stimulation device was manufactured by VMT Co., Ltd. (VMT, Pohang-si, Republic of Korea). This aging accelerator was designed to use 220 V or less, with either alternative current or direct current to be used selectively (Figure 1). The electrode was directly inserted into the muscle through the carcass backfat. Then, localized electrical stimulation treatment was applied.

### 2.3. pH and Color

pH was determined using a pH meter (Model203, Testo, Lenzkirch, Germany) with a temperature sensor and a glass probe that was directly inserted into meat samples. The pH meter was standardized with standard solutions of pH 4, 7, and 9 prior to measurement. Meat color was measured at three different areas on the surface after 30 min of blooming at 4 °C in a dark environment using a color meter (CM-2500d, Konica Minolta, Tokyo, Japan) with a D65 illuminator, an angle of 10°, and a diameter of 8 mm. Their mean value was used for analysis. Meat color was expressed as CIE L* (brightness), a* (redness), and b* (yellowness). Before each measurement, the apparatus was calibrated using a white ceramic tile (Minolta calibration plate, Y = 84.2, x = 0.3197, y = 0.3367).

### 2.4. Chemical Composition

Chemical composition was analyzed following the AOAC method [22]. Moisture content (AOAC no. 950.46) was measured using a Halogen moisture analyzer (HR73, Mettler Toledo, Greifeness, Switzerland). A 2.5 g sample was spread thinly on an aluminum dish to use with a moisture meter and then dried at 105 °C. When all the moisture was dried, the moisture content was automatically calculated. An average value was used by repeating this measurement twice. To measure crude fat content (AOAC no. 969.39), 5 g of each sample was put into a cylindrical filter (Whatman#2800-432), and 1.5 g of sea sand was mixed with the sample. The prepared sample was completely dried in a dry oven (DryG-SL105, DAIHAN Science, Wonju-si, Republic of Korea) at 102 °C for 5 h, then taken out and cooled in a desiccator for 30 min. The cooled sample/cylindrical paper was placed in a Soxhlet connected to a round-bottom flask containing 120 mL petroleum ether, and fat was completely extracted for 6 h by setting the temperature of the water bath for refluxing in the Soxhlet to 100 °C. In the 100 °C water bath, all petroleum was evaporated in a round-bottom flask that contained the extracted petroleum, then dried in a dryer at 102 °C for 1 h, after which it was cooled in a desiccator for 30 min. The weight of the extracted crude fat was calculated as the increase in the weight of the round-bottom flask before and after extraction, and the crude fat content was expressed as a percentage of the sample weight.

### 2.5. 2-Thiobarbituric Acid Reactive Substances (TBARS) Value

Lipid oxidation was assessed by the TBARS method [23]. First, 2.5 g of the beef sample was mixed with 7.5 mL of distilled water and 10 mL of TBA-TCA solution (0.29% thiobarbituric acid, 15% trichloroacetic acid). The mixture was homogenized for 15 s at 11,000 rpm using a homogenizer (T25b, IKa Werke Gmbh & Co., Breisgau, Germany), to which 25 μL of BHA (butylated hydroxyanisole) was added to prevent further oxidation. The homogenate was filled up to 30 mL with distilled water, and then, the 30 mL of homogenate was heated at 90 °C for 15 min in a water bath. After heating, the homogenate was immediately cooled for 20 min in an ice bath, followed by centrifugation (3000 rpm, 10 min, 4 °C) using a centrifuge (X-30, Beckman Coulter, Brea, CA, USA). The upper layer (1 mL) was taken, and its absorbance was measured at 531 nm using a UV spectrophotometer (Ultrospec2000, Pharmacia Biotech, Cambridge, UK). The results are expressed as mg malondialdehyde/kg meat.
TBARS (MA mg/kg) = Absorbance × 5.88

### 2.6. Cooking Loss and Warner–Bratzler Shear Force (WBsf)

Cooking loss and WBsf were analyzed at day 2 or day 14 of aging for the samples stored at 4 °C. Each meat sample had fat and connective tissue removed from the exterior of the 2.5 cm thick sample before weighing. After that, a digital temperature recorder (TR-52, T&D Corporation, Matsumoto, Japan) sensor was inserted into the center of each steak, and then, each one was placed in a polyethylene bag (LLDPE/LDPE/LDPE/Nylon 4-layer film material vacuum bag) to prepare for heating. Each sample was heated in a constant-temperature water bath (SH.SWB15, DAEHAN Sci., Wonju-si, Republic of Korea) at 70 °C until the core temperature of each reached 70 °C, and then cooled in running water at about 18 °C for 30 min. The weight of each was measured after removing water from the cooled sample using paper towel. The weight was calculated as a percentage using the weight before/after heating. After measuring the heating loss, six or more cores with a diameter of 1.25 cm were collected using a metal bore (diameter 1.23 cm) parallel with the muscle fiber of the sample, and shear force was measured using an Instron (Model 4465, Instron, Norwood, MA, USA). A 50 kg load cell was used with a crosshead speed of 400 mm/min. Shear force was measured using a V-shaped blade to cut at right angles to the muscle fibers, and the maximum shear force (kg) was measured.

### 2.7. Calculation of CO_2_-eq Emission Mitigation Achieved by Reducing the Fattening Period

Table 1 presents the details used to estimate the effect on carbon emissions of shortening the fattening duration of Hanwoo cattle. The amount of CO_2_ generated from 1 kg of trimmed meat for each fattening period (26 and 31 months) was calculated as follows:Trimmed meat amount per 1 kg of live cattle (kg) = (D × B)/100(1)
Amount of CO_2_ emitted per 1 kg of the trimmed beef (CO_2_-eq) = F × (1)(2)
Amount of CO_2_ emitted per 1 kg of the trimmed beef (CO_2_-eq) for the additional 5 months of feeding period = (2) of 31 months − (2) of 26 months

### 2.8. Statistical Analysis

The statistical analysis was conducted using the mixed model of IBM SPSS Statistics (version 27.0, SPSS Inc., Chicago, IL, USA) with fixed effects of localized electrical stimulation and aging and random effect of the carcass. The effect of localized electrical stimulation and aging was analyzed by two-way analysis of variance (ANOVA), and deviation in WBsf value due to localized electrical stimulation was described in the 95% confidence interval (95% CI).

## 3. Results

### 3.1. Effect of Localized Electrical Stimulation on Hanwoo Beef Quality

The effect of localized electrical stimulation and aging on the chemical properties of the LT and BF muscles of the Hanwoo steer is detailed in Table 2. No significant differences in pH resulted from localized electrical stimulation. The pH did decrease during the aging period, but it remained within a normal range for beef [15,25]. L*, a*, and b* of the localized electrical stimulation group at aging day 2 and 14 were mostly higher than in the control, except the b* of the BF muscle, but there was no significant electrical stimulation effect. There was a significant difference in aging effect on a* and b*. There were no significant differences in the moisture and fat contents between the two muscles on day 2 of aging regardless of localized electrical stimulation or control treatments. There was also no significant effect of localized electrical stimulation and aging treatment on the levels in both muscles. Furthermore, we could not find any interactive effect between electrical stimulation and aging treatments on the chemical properties of the Hanwoo beef LT and BF muscles.

Table 3 shows the effect of localized electrical stimulation and aging on the physical properties of the LT and BF muscles from Hanwoo beef. No effect on drip loss and cooking loss was observed in the LT and BF muscles after localized electrical stimulation. In both muscles, the effect of reducing shear force due to aging treatment and electrical stimulation treatment was significant. In the LT muscle, the shear force-reduction effect due to aging was found to be greater than the shear force reduction effect due to localized electrical stimulation, but in the BF muscle, both localized electrical stimulation and aging treatments were found to have a highly significant effect. However, there was no interactive effect between electrical stimulation and aging treatments on the WBsf of the Hanwoo beef LT and BF muscles. The measured effect of localized electrical stimulation on the reduction in tenderness deviation by the LT and BF muscles is shown in Figure 2. The localized electrical stimulation treatment at the early phases of the aging process decreased the tenderness deviation in both the LT and BF muscles, and the effect of this was clearly apparent in the BF muscle. After 14 days of aging, the tenderness deviation in the BF muscle was greatly reduced.

### 3.2. Effect of CO_2_ Emission Reduction from Hanwoo Meat through Localized Electrical Stimulation

Table 4 shows the effect of localized electrical stimulation of the carcass on improving the tenderness of the LT and BF muscles on aging days 2 and 14. Based on the shear force value of the LT muscle at day 2 in the control, the effect of reducing the shear force value of these by aging was greater than that of localized electrical stimulation. For the BF muscle, the initial shear force decreased by 1.5 kg, which was greater than the decrease in shear force (−1.27 kg) observed in the control group after 14 days, indicating that the starting point of tenderization by aging for the BF muscle could be lower than the WBsf of the control group at 14 days, or even more.

Table 5 shows the estimated CO_2_ emissions from 1 kg of trimmed Hanwoo beef based on the trimmed-meat yield for each fattening month using the results of these previous studies. One kilogram of trimmed Hanwoo beef produced from a 26-month-old castrated cow generated 24.56 kg of CO_2_-eq, and 25.60 kg CO_2_ was emitted by 1 kg of trimmed Hanwoo beef at 31 months of age. As a result, 1 kg of CO_2_ was additionally generated per 1 kg of beef during an additional 5 months after 26 months of age. In other words, 5 months of additional rearing is required to reduce the shear force by about 1 kg through long-term fattening [9]; it could be estimated that an additional 1.04 kg of CO_2_ is emitted per 1 kg of finished beef during this additional rearing period.

## 4. Discussion

Electrical stimulation at the beginning of slaughter physically breaks down muscle fibers [15,26]. Meat from animals with a high muscle fiber density and low intramuscular fat, such as the BF muscle in beef, may receive greater benefits from electrical stimulation applied during the early stages of slaughter, where it can trigger a greater amount of physical breakdown in muscle structure [27]. The strong contractions generated during the electrical stimulation treatment weaken the muscle structure, allowing oxygen to penetrate deeply into the cut surface, resulting in a thicker oxymyoglobin layer than that of beef that did not receive the same treatment [26,28]. As the structure of the BF muscle, which has a higher density of muscle bundles and less fat than the LT muscle [29], might be weakened by localized electrical stimulation, BF showed a significantly higher a* value at day 14 of aging as muscle fiber bundle breakdown was accelerated. There were no significant differences in the moisture and fat contents between the two muscles on day 2 of aging regardless of localized electrical stimulation or control treatment. There were also no significant differences in TBARS levels between the two muscles at either day 2 or day 14 of aging. The electrical stimulation did not affect the TBARS value of beef in conjunction with ES and storage duration [30]. However, some studies have shown increased lipid oxidation in alpaca meat [31]. In our research, there was no significant lipid oxidation due to localized electrical stimulation treatment. These results indicate that electrical stimulation of the carcass has no significant impact on chemical stability or meat storage stability. In previous research findings, electrical stimulation significantly decreased the early shear force during maturation [30], and its effect on tenderizing meat persisted even after 14 days [32]. At aging day 2, the WBsf value of the localized electrical stimulation group was lower than the control group in both the LT and BF muscles. However, at aging day 14, the WBsf value in the localized electrical stimulation group for the BF muscle was only lower than the control. Both electrical stimulation and aging are methods that improve meat tenderness [12], but research comparing the impact strength of their effects on different muscles has not yet been conducted. In this study, the LT muscle showed significant effects in response to electrical stimulation in the early stages of maturation, whereas after 14 days, only the improvement in tenderness due to maturation remained. There were significant impacts from both localized electrical stimulation and aging treatments on the WBsf value of the LT and BF muscles, but no interaction between localized electrical stimulation and aging was found. In particular, the WBsf value at aging day 2 of the electrically stimulated BF muscle was 5.35 kg, which was lower than that of the control group (5.58 kg) after 14 days of aging. On the other hand, the effect of WBsf reduction due to aging in the LT muscle was higher than the localized electrical stimulation effect. These results were expected to be caused by differences in the effects of electrical stimulation depending on the degree of intramuscular fat.

Consumers prefer tenderloin to outside round [25]. When purchasing beef, they consider tenderness the top priority [33]. The WBsf range that consumers perceive as tender is 3.2–3.9 kg [34]. For grade-1-quality Korean beef tenderloin to reach the WBsf value of tender beef required about 2 days of aging, while the outside round takes 28 days or more of aging [17]. The fat contents of grade-1-quality Hanwoo beef tenderloin and outside round are similar (11.26% and 11.0%), but the shear forces of tenderloin and outside round are 2.98 and 5.15 kg [35]. To summarize, consumers may avoid cuts like beef round not because of the intramuscular fat content, but rather because these cuts are less tender. In South Korea, most Hanwoo steers are typically fattened for an average of 30.7 months to produce highly marbled beef [36]. When calculating the effect of increasing intramuscular fat and decreasing shear force through the long-term fattening of steer, an additional fattening period of approximately 5 months might increase intramuscular fat by 4.23% and decrease shear force by 0.98 kg [10]. According to a report in 2005 by the Korean Rural Development Administration [10], additional feeding carried out for 5 months after 26 months of age resulted in an increase in the live weight of Korean cattle of about 120 kg, but the commercial trimmed meat yield decreased by about 2.7%. Moreover, the longer-term fattening of Hanwoo cattle (additional fattening for 5 months) increased intramuscular fat by 4.32%, decreased shear force by 1 kg, and ultimately increased CO_2_ emissions by 1 kg in relation to the shorter fattening period (Table 5). This was similar to the case of Japanese Wagyu beef, which was raised using a beef production system that allowed for the accumulation of high levels of intramuscular fat, and about 1 kg CO_2_-eq of GHG emissions was emitted per kilogram of beef over a period of 2 months [11]. As a result of our experiment, the shear force at aging day 2 was reduced by 0.59 and 1.50 kg of shear force (LT and BF, respectively) in 31-month-old Korean steers through localized electrical stimulation (Table 4). Moreover, the shear force of the BF muscle at aging day 14 (−2.48 kg) was lower than that at aging day 2 (−1.50 kg), which suggests that ES facilitated the aging process and decreased the number of aging days required for tenderization in relation to the control group (−1.27 kg after 14 days for BF muscle). When 1 kg of meat is aged (or stored) for a single day under supermarket storage conditions, 0.0009 kW of electricity is used [37], and the carbon emissions associated with electricity consumption in South Korea are 0.495 CO_2_-eq/kWa [38]. Considering that the average amount of trimmed meat obtained from a 26-month-old Hanwoo steer is 259 kg, it is estimated that an additional 0.64 kg of CO_2_-eq is generated after aging a Hanwoo steer for 14 days. In other words, shortening the aging period to achieve the tenderness desired by consumers could potentially mean that carbon emissions additionally emitted by aging can be reduced. This suggests that lowering the shear force through localized electrical stimulation after slaughter may be one way to reduce the environmental burden of the beef industry, rather than using long-term fattening of Hanwoo steer.

In this experiment, we estimated the effect of localized electrical stimulation on reducing shear force and carbon emissions per functional unit in 31-month-old Hanwoo steers, but the effect of localized electrical stimulation is expected to be greater in Hanwoo steer with less intramuscular fat (or fewer shipping months). This suggests the need for additional research in this field. In addition, this study only calculated GHG emissions using electricity consumption, and once refrigerant gas use and storage space are factored in, the environmental impact may be greater than we anticipated. We suggest that a comprehensive evaluation of GHG emissions should be undertaken in the post-farm gate stage of beef production in Korea.

## 5. Conclusions

Based on the results of our study, localized electrical stimulation had no significant effect on the chemical stability of LT and BF muscles but decreased their initial shear force. In particular, the BF muscle (with low intramuscular fat) treated with ES had a lower shear force value in the treatment group at day 2 of aging than in the control group at day 14 of aging. As shown in this study, by shortening the fattening period with high-energy specifications in the late stages of beef production, which increases the intramuscular fat content, and by lowering the initial shear force through localized electrical stimulation, it is possible to produce beef with less intramuscular fat but improved tenderness and a lower carbon footprint. As there is high demand for low-fat meat from health-conscious consumers, we expect that this method of producing healthy meat with low intramuscular fat and good flavor can establish itself as a new approach to sustainable meat production and consumption.

## Figures and Tables

**Figure 1 foods-13-00037-f001:**
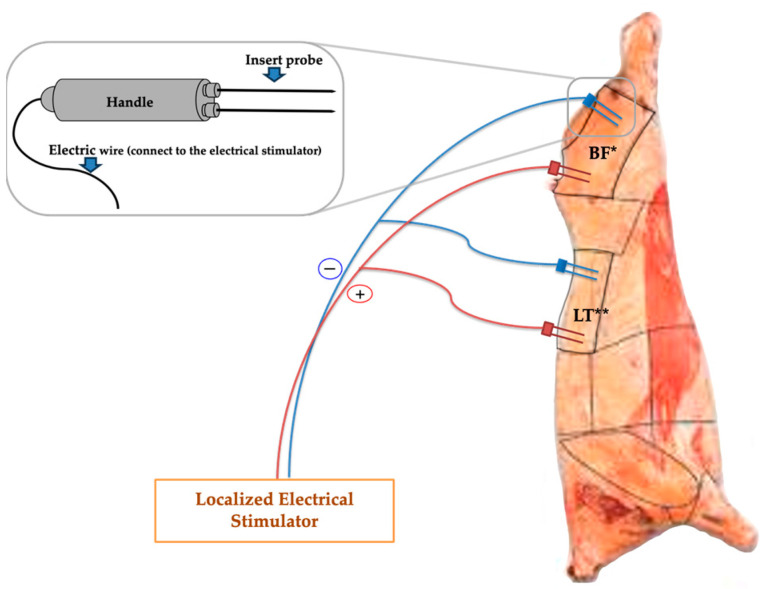
Localized electrical stimulation probe and insertion locations. * BF: Biceps femoris muscle; ** LT: Longissimus thoracis muscle.

**Figure 2 foods-13-00037-f002:**
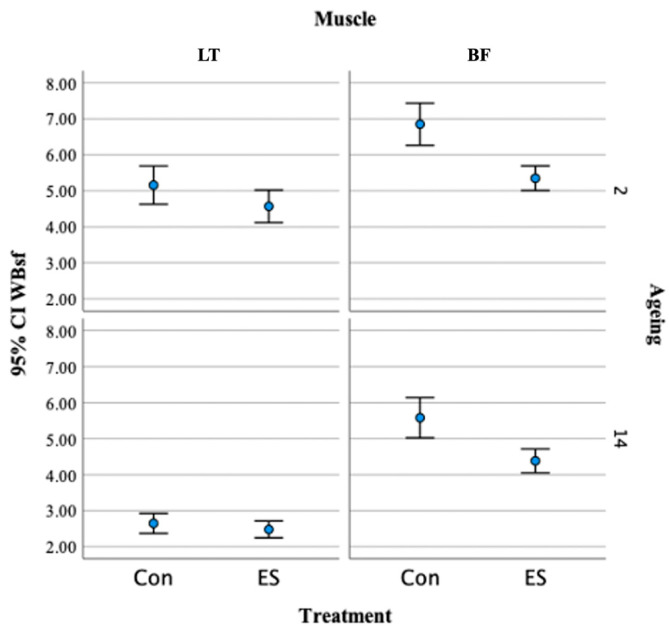
Effect of localized electrical stimulation on the reduction in tenderness deviation in LT and BF muscles.

**Table 1 foods-13-00037-t001:** Estimated changes in intramuscular fat content, WBsf of beef, and CO_2_ emissions of Hanwoo steer using different levels of fattening duration.

Properties of a Hanwoo Cattle/Carcass/Beef	Fattening Duration			References
	26 Months	31 Months	31–26 Months	
A. Average intramuscular fat content (%) *	14.83	18.61	4.23	[10]
B. Average live weight (kg) **	655.9	776.8	121.0	[10]
C. Amount of the trimmed meat (kg) ***	259.65	321.98	62.33	[10]
D. Trimmed meat yield (%) ***	67.37	64.63	−2.74	[10]
E. WBsf of beef (kg of shear force) *	4.44	3.46	−0.98	[10]
**GHG emissions**	**CO_2_-eq**	**Functional unit**		
F. Hanwoo cattle	16.55	Kg of live weight		[24]

* Value of loin, ** Of the cattle, *** From/of a carcass.

**Table 2 foods-13-00037-t002:** Effect of localized electrical stimulation and aging on the chemical properties of Hanwoo beef LT and BF muscles.

Muscle		2 Days Aging		14 Days Aging		*p* Value		
		Control	ES ^1^	Control	ES	ES	Aging	Interaction
Longissimus thoracis								
pH		5.53	5.53	5.50	5.50	0.888	0.003	0.844
Meat color	CIE L*	39.63	41.68	40.8	41.3	0.129	0.636	0.353
	CIE a*	16.69	17.39	20.54	20.93	0.415	<0.001	0.819
	CIE b*	13.18	14.14	16.31	16.97	0.059	<0.001	0.725
Moisture (%)		56.47	54.71	-	-	0.360	-	-
Crude fat (%)		23.79	21.81	-	-	0.465	-	-
TBARS (MA mg/kg)		0.26	0.31	0.31	0.32	0.194	0.155	0.323
Biceps femoris								
pH		5.53	5.56	5.50	5.52	0.117	0.003	0.618
Meat color	CIE L*	37.79	38.82	38.03	38.57	0.147	0.989	0.656
	CIE a*	19.33	20.84	23.38	25.69	0.012	<0.001	0.588
	CIE b*	13.59	15.01	16.24	17.88	<0.001	<0001	0.785
Moisture (%)		61.30	61.69	-	-	0.826	-	-
Crude fat (%)		19.7	18.48	-	-	0.655	-	-
TBARS (MA mg/kg)		0.36	0.38	0.36	0.41	0.303	0.673	0.683
Df ^2^			3/79					

^1^ ES: Localized electrical stimulation. ^2^ Df: Degree of freedom.

**Table 3 foods-13-00037-t003:** Effect of localized electrical stimulation and aging on the physical properties of Hanwoo beef LT and BF muscles.

Muscle	2 Days Aging		14 Days Aging		*p* Value		
	Control	ES ^1^	Control	ES	ES	Aging	Interaction
Longissimus thoracis							
Drip loss (%)	-	-	20.92	16.79	0.346	-	-
Cooking loss (%)	13.79	15.48	14.26	15.57	0.021	0.660	0.765
WBsf (kg)	5.16	4.57	2.65	2.48	0.047	<0.001	0.267
Biceps femoris							
Drip loss (%)	-	-	7.40	7.34	0.950	-	-
Cooking loss (%)	18.21	19.05	23.75	23.76	0.528	<0.001	0.532
WBsf (kg)	6.85	5.35	5.58	4.83	<0.001	<0.001	0.498
Df ^2^		3/79					

^1^ ES: Localized electrical stimulation. ^2^ Df: Degree of freedom

**Table 4 foods-13-00037-t004:** Effects of localized electrical stimulation of the carcass on the improving tenderness of LT and BF muscles on aging days 2 and 14.

Aging Days	Change in WBsf (kg) *			
	LT Muscle		BF Muscle	
	Control	ES	Control	ES
2	0 *	−0.59	0 *	−1.50
14	−2.51	−2.68	−1.27	−2.48

* Calculated based on 0 points of change in WBsf by both electrical stimulation treatment and aging on each muscle.

**Table 5 foods-13-00037-t005:** Estimated CO_2_ emissions from the production of 1 kg of trimmed Hanwoo beef with different feeding periods.

	Feeding Period		
	26 Months	31 Months	Additional 5 Months *
Amount of CO_2_ emissions per 1 kg of the trimmed Hanwoo beef (CO_2_-eq)	24.56	25.6	1.04

* Calculated using differences in CO_2_-eq emissions between 31 and 26 months.

## Data Availability

Data is contained within the article.

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
