# Peer review of "Beef Toughness and the Amount of Greenhouse Gas Emissions as a Function of Localized Electrical Stimulation"

_foods, 2023, doi:10.3390/foods13010037_

Round 1

Reviewer 1 Report (Previous Reviewer 3)

Comments and Suggestions for Authors

There continue to be language difficulties that would be easily corrected by a native English speaker.

Line(s)      Comment

15, 17       Abbreviations and acronyms should be defined at first use.

22-24        Poorly grammar.

24-27        Some of the information is redundant with that in l. 16-20.

28-30        Poor grammar; “on Hanwoo steer was shortening the fattening”.

34-35        Poor grammar; “stimulation improved tenderness”.

37-38        It is inappropriate to indicate that the results meet the demands of health-conscious consumers or the meat reduces carbon footprint since consumer demands and the actual carbon footprint were not measured in this study.

43-44        Reference(s) needed to substantiate that the information in this sentence is factual.

44-46        The 2006 FAO report has repeatedly been shown to be inaccurate, even by FAO, so the newest FAO documents on greenhouse gas emissions should be cited.

48-50        Reference(s) needed to substantiate that it is necessary to develop carbon reduction technologies in Korea.

53-55        The meaning of this sentence is unclear.

61-62        Reference(s) needed to substantiate that efforts to reduce greenhouse gas emissions in the post-farm gate stage are inadequate.

79-83        Run-on sentence.

90-91        This is repetitive of l. 71-72.

92-93        Reference(s) needed to substantiate that common electrical stimulation is performed on the whole carcass at slaughter.

98              “a possibility of shortening the fattening”

104-105   “composing three” or “composed of three”

106-107   Some of the slaughter practices, including time and distance of transport, time in lairage before slaughter, and immobilization method and the temperature and humidity must be given as these factors will affect the postmortem metabolic activities that influence tenderness and lipid stability.

128           The word “stably” is not understood.

179-183   “Each” indicates “one,” so “each meat sample had fat and connective tissue removed from the exterior of the 1.5-cm thick sample before weighing the sample. After that a digital temperature recorder (TR-52, T&D Corporation, Japan) sensor was inserted into the center of the steak, which was”

204           “E. WBSF of beef”

209-212   A block design and analysis must be used because the use of one side for electrical stimulation and the other side as a control (l. 111-112) on each carcass limits the statistical inferences that can be made since the treatments were not randomized among the two sides of each carcass.

226-228   Poor language. “stimulation effect. There was significant aging effect on a* and b*.”

229           “fat contents between”

230           “regardless of electrical stimulation or control treatments.”

243           “Table 3 shows the”

264           “muscle was greatly reduced.”

314-315   The meaning of “trading meat yield” and “of each shipping month” are not clear.

340           The breakdown of muscle fibers was not measured in this study so “physically breaks down muscle fibers,”

349           The muscle structure weakness was not measured in the present study so “treatment weakens the muscle”

356-357   “regardless of electrical stimulation or control treatment.”

360           “meat storage stability. The WBSF of”

360-362   Poorly worded and run-on sentence.

363-364   Improper grammar.

370-374   Poorly worded and run-on sentence.

383-384   Poor wording.

393-396   Meaning of the sentence is not clear.

402           “aging day 2 was reduced”

406-409   Meaning of the sentence is not clear.

454-456   Poorly worded.

463           Delete “be”

Comments on the Quality of English Language

The reputation of a journal is based both on the scientific contributions and the presentation of the information. There are continuing language difficulties in the manuscript. Only some of the previous reviewer comments are adequately addressed in the most recent manuscript version. The manuscript should be rejected to maintain the integrity and reputation of the journal.

Author Response

Reviewer 2 Report (Previous Reviewer 2)

Comments and Suggestions for Authors

Dear Editor and Authors,

I have read the corrected version of the paper “Greenhouse gas emission and reduction of bovine toughness by localized electrical stimulation” such as “Beef toughness and amount of greenhouse gases emission as a function of localized electrical stimulation” and have some comments. First of all, the Authors did a good job analysing the results using two-way ANOVA and showing the effect of ES and ageing. Nevertheless, I advise the Authors to take a more critical point of view when analysing the results about the impact of ES on beef tenderness (in my opinion the effect although statistically significant is not great and ES does not enable to reduce the ageing time of LTL and BF (at least in the case of LTL in this experiment design).  

The part about using ES and ageing is generally well written, however, I still don’t understand the sentence about shortening the ageing of BF for more than 14 days… lines 302-304 “For the BF muscle, initial shear force decreased by 1.5 kg, which was greater than the decrease in shear force (-1.27 kg) observed in the control group after 14 days, indicating that the ageing period for the BF muscle could be made shorter by 14 days, or even more” why shorter? I would say that even though ES was used the BF meat was not tender (therefore the information about shortening the ageing period is not true and might be harmful to the beef quality if applied in the practice). It might be concluded that even ES combined with 14d ageing was not enough to produce tender BF meat from 26-months-old animals. Please note that even when ES was used the WBSF was higher than required for tender beef (3.2 kg – 3.9 kg – as stated in line 372) after 2 and 14d-ageing. Please bear in mind when revising the Abstract and conclusions as well, i.e. line 454 “… significantly decreased the initial shear-force of both of LT and BF muscles, especially BF muscle with low intramuscular fat shown lower level than the 14 days ageing shear force of the control”.

I am still confused by the part about CO2 emissions. First of all, if the Editor agrees, I suggest making a separate part of the manuscript – i.e. in the discussion, solely dedicated to CO2 and gathering there all the information about CO2 emissions from other sections of the manuscript (materials and method and results). This will make the part about CO2 emissions more concise and easy to follow. The equation for calculations of CO2 emissions was corrected, but still, there is no formula, which was used by the Authors to calculate the values enclosed in Table 5. Please provide the equation to make it possible for readers to follow your train of thought.

Line 385-387 I assume that there is a mistake – please consult - there are 4.32% and 4.23% “an additional fattening period of approximately 5 months might increase intramuscular fat by 4.32 % and decrease shear force by 0.98 kg [9]. In other words, an increase in fat content of 4.23 % is required to lower the shear force by about 1 kg”. Please add a discussion if a reduction in WBSF of 1 kg is much or not. Would it result in tender beef or not, etc.?

Line 409-411 “In other words, shortening the ageing period to reach the tenderness desired by consumers can be interpreted to mean that carbon emissions that may be additionally emitted by ageing can be reduced. This suggests that lowering the shear force through local electrical stimulation after slaughter may be one of the ways to reduce the…” Generally, it is true that reduced ageing lowers CO2 emissions, but I don’t agree with the statement that it is possible to shorten the ageing of beef by using ES based on the results of this study. The values of WBSF reported for electrostimulated 2d aged beef (LT and BF) were much higher than those for tender beef, so based on the results the statement about shortening ageing thanks to ES is not true. Please revise and rewrite.  

Line 441- “After estimating the effects of localized electrical stimulation on carbon dioxide emissions in Hanwoo steer, we concluded that electrical stimulation had little effect on the chemical stability of the LT and BF muscles” corrects into “After estimating the effects of localized electrical stimulation on beef from Hanwoo steer, we concluded that electrical stimulation had little effect on the chemical stability of the LT and BF muscles”

458-460 “This suggests that the production of tender beef with a  shorter fattening period and lower intramuscular fat, and therefore a lower carbon footprint, can be achieved by shortening the fattening period of high-energy specifications in the late stages of beef production and accelerating ageing with electrical stimulation” – please note that you obtain a tender beef only from the tenderloin, not BF; what do you mean by “accelerating ageing”? From the results it is obvious the ageing should be 14d for tenderloin and even longer for BF than 14d. The reduction of WBSF in tenderloin in 14d was 0.17 kg and 0.75 kg for BF. You may discuss if it is much taking into account energy consumption and CO2 emissions.

Author Response

Reviewer 3 Report (Previous Reviewer 1)

Comments and Suggestions for Authors

Dear authors, the revised manuscript is interesting, however, the following changes need to be made:

Line 13: insert space… 45 V

Line 13: indicate the meaning of the abbreviation TBARS, or only include the words lipid oxidation

Line 14: include the meaning of the abbreviation WBsf

Line 18: interaction

Line 19: insert space… 5.35 kg

Line 19: insert space… 5.58 kg

Line 61: The abbreviation GHG must be used from the first time it appears in this section, and subsequently, only use the abbreviation throughout the document where it is required.

Line 81: insert space… methods [11],

Line 82: insert space… countries [13].

Line 86: modify… improvement [36],

Line 86: carbon dioxide (insert abbreviature and used it through the manuscript were it is required)

Line 93: Animals and Treatment

Line 103: use 1 min instead of 60 sec

Line 104: use h instead of hr

Line 105,106: use only the abbreviation for muscles, this has already been described previously

Line 107: use h instead of hr

Line 113: Ageing Speeder through Localized Electrical Stimulation

Line 123: pH and Color

Line 123: Is information about pH and color results included in the manuscript abstract?

Line 124-132: insert the references of the methodologies used in measuring pH and color

Line 130: (yellowness).

Line 132: 0.3367).

Line 134: Chemical Composition

Line 138: moisture meter (insert equipment information)

Line 142: dryer (insert equipment information)

Line 143: did you mean 5 h?

Line 145: use h instead of hr

Line 148: use h instead of hr

Line 153: Thiobarbituric Acid Reactive Substances (TBARS)

Line 154: by TBARS method

Line 161: centrifuge (insert equipment information)

Line 167: Cooking Loss Weight (CLW) and Warner Bratzler Shear Force (WBSF)

Line 168: use CLW and WBSF abbreviatures through the manuscript

Line 168-183: What are the references of the methods used in this section, CLW and WBSF?

Line 185: Calculation of CO2-eq Emission Mitigation that could be Achieved by Reducing the Fattening Period

Line 188: It is necessary to use abbreviations after the first time they are mentioned

Line 193: insert spaces between the number and the word months

Line 193: use 26-31 months instead of 31-26months

Line 218: in table content, did you mean mg MA/kg

Line 219: Why not also use the meaning and the abbreviation (ES) in the abstract and in the introduction, as well as only the abbreviation in the document after being indicated in the introduction

Line 226: LD? Or LT?

Line 244: Effect of CO2 Emission Reduction from Hanwoo Meat through Localized Electrical Stimulation

Line 257,263: 1 kg

Line 264: 1.04 kg

Line 264,266: CO2

Line 298: part [23].

Line 299: beef [28],

Line 300: 3.2 to 3.9 kg

Line 307: beef [31].

Line 319: beef (Table 5).

Line 324: (Table 4).

Line 331: CO2-eq

Line 349: What is little effect? some percentage? Does it have an effect or not?

Line 386: the abbreviated name of the journal must be in italic text format

Line 384: 1721-1736

Line 384: use italic text format for 27(9)

Line 389: use the abbreviated format for the reference journal

Line 393: 2115-2122.

Line 398: 451-455

Line 404: 167-174

Line 406: In this title, uppercase and lowercase words are mixed, is this correct according to the authors' guide?

Line 412: 281-288

Line 414: 929-941

Line 420: P10098-10103

Line 423: 427-440

Line 424: In this title, uppercase and lowercase words are mixed, is this correct according to the authors' guide?

Line 425: 859-866

Line 431: In this title, uppercase and lowercase words are mixed, is this correct according to the authors' guide?

Line 432: 479-483

Line 436: 236-242

Line 441: 357-364

Round 2

Reviewer 1 Report (Previous Reviewer 3)

Comments and Suggestions for Authors

There continue to be language difficulties that would be easily corrected by a native English speaker. The incorrect experimental design and statistical analyses of the data require rejection of the manuscript.

Line(s)      Comment

14-19        “Localized electrical stimulation and aging treatments had a significant effect on meat tenderness in LT and BF muscles, but there was no interactive effect.”

25              Delete “was shortening the fattening period of Hanwoo beef production”

30-32        Repetitive of l. 23-24.

35              “and reduced CO2

163           Delete “oven” duplicate word.

227-228   The statistical analysis is incorrect. The experimental unit is the smallest entity to which a given treatment is applied. Localized electrical stimulation was only on the right carcass side muscles (l. 117-118), which restricts randomization of the treatment. Aging was done on muscles from both sides of the carcass. A split-plot or block experimental design must be used to analyze the data.

276, 286, 325 This is incorrect. Only muscles on one side of the carcass received localized electrical stimulation.

320-340   It was expected that the results of electrical stimulation and ageing from this study would be compared to results from other studies using these treatments on beef.

402           “stimulation had no effect on”

404           “both LT and”

404           “fat had lower”

Comments on the Quality of English Language

Most of the revised sections of the manuscript had language difficulties.

Author Response

Reviewer 2 Report (Previous Reviewer 2)

Comments and Suggestions for Authors

The paper was corrected. I have no further comments. 

Author Response

This manuscript is a resubmission of an earlier submission. The following is a list of the peer review reports and author responses from that submission.

Round 1

Reviewer 1 Report

Comments and Suggestions for Authors

The information obtained in this investigation is interesting, however, it is necessary to attend to the following:

Line 1: remove parenthesis

Line 7: insert a comma like in line 5.… Jeonju, 55069

Line 14: insert space…. over 14 days

Line 18: modify… stimulation; … ageing;….tenderness; CO2 emision;

Line 34: What is the meaning of GHG?

Line 43: delete parenthesis

Line 51: use italic text format for post-mortem

Line 56: insert space… ageing [11]

Line 57: insert space… meat [12].

Line 57: delete… by Park et al.

Line 60: insert space… 3.2 kg

Line 61: delete… Claeys et al.

Line 61: modify… Another authors [14] reported

Line 63: modify… Also, it has been reported [15] that aging

Line 76: remove indent space

Line 76: use italic text format for the subsection

Line 76: 2.1.

Line 81: Longissimus         

Line 86: insert space…. 45 V

Line 86: Why are time units sometimes abbreviated and not others? it is necessary to standardize through the document

Line 89: 36 hrs, 36 hours or 36 h?

Line 89: postmortem or post-mortem like previously described: italic text format?

Line 92: insert space…. 4 °C

Line 94: remove indent space

Line 94: use italic text format for the subsection

Line 94: 2.2.

Line 96: insert space…. 220 V

Line 101: Increase the font size of the information contained within the figure, the font is distorted as the size of the image increases.

Line 101: Is it possible to improve the resolution quality of the carcass image?

Line 105: remove indent space

Line 105: use italic text format for the subsection

Line 105: 2.3.

Line 106: delete repeated parenthesis

Line 110: insert space…. 4 °C

Line 105-112: What were the methods used to determine pH and color coordinates?

Line 111: in color determination, what was the degree of observer used and what type of calibration was used?

Line 114: remove indent space

Line 114: use italic text format for the subsection

Line 114: 2.4.

Line 115-130: indicate the AOAC method number for each determination

Line 115-130: the reference was missed

Line 118: insert information about the equipment used (model, trademark, country)

Line 118,121: delete space… °C

Line 120: filter paper # ???

Line 124: 120 mL

Line 125: insert space…. 100 °C

Line 126: delete space… °C

Line 132: remove indent space

Line 132: use italic text format for the subsection

Line 132: 2.5.

Line 133: by TBARS method [17].

Line 139: 30 mL

Line 129: insert space…. 90 °C

Line 141: insert space…. 4 °C

Line 142: insert the trademark of the equipment

Line 144:  TBARS (MA…

Line 144: Absorbance

Line 146: remove indent space

Line 146: use italic text format for the subsection

Line 146: 2.6.

Line 148: 4C or 4 °C?

Line 151,152: 70 °C

Line 151: insert information about the equipment used (model, trademark, country)

Line 152: 18 °C

Line 160: remove indent space

Line 160: use italic text format for the subsection

Line 160: 2.7.

Line 164: insert space…. 1 kg

Line 164: as follows:

Line 164,165: Is the information contained in the table equations? If it is correct, it is necessary to use the format for the equations described in the Foods Authors' Guide, you can consult the Microsoft word template

Line 166: remove the bold text format for the table title

Line 167: insert a dot at the end of the table title

Line 167: in the information contained in the table insert spaces to 26 months, etc

Line 169: remove indent space

Line 169: use italic text format for the subsection

Line 169: 2.8.

Line 177: remove indent space

Line 177: use italic text format for the subsection

Line 177: 3.1.

Line 181: modify… for beef. L*, a* and b* of both….color coordinates were previously abbreviated, it is not necessary to repeat the non-abbreviated form

Line 181-183: Respect "Lightness, redness, and yellowness of both muscles at ageing day 2 (after localized electrical stimulation) were slightly higher than in the control, " how can they be slightly higher if they share the same literal ?

Line 183: except in b* values of the BF

Line 184: day14, L* and a* values of the

Line 184: Respect "lightness and redness of the BF muscle were significantly higher than control". In the information contained in the table, the values do not have literals, which is associated with the level of significance, and on the contrary, it is inferring that the values of L* are slightly higher than the control

Line 185-188: information should not be discussed in the results section

Line 191: significantly high a* values at

Line 193: on day 2 and 14 ?

Line 197: remove the bold text format for the table title

Line 198: insert a dot at the end of the table title

Line 198: in table content correct…Crude fat…mg/kg

Line 198: Is the measurement of color a chemical or physicochemical property?

Line 210: insert space…. 5.35 kg

Line 211: insert space…. 5.58 kg

Line 221: insert a dot at the end of the table title

Line 221: the format of the information contained in the upper part of the table is different from that of the previous table, it is necessary to homogenize

Line 229: insert a dot at the end of the figure title

Line 230: delete space

Line 232,238,239,241,243: information should not be discussed in the results section. Note: Did you mean on line 176, Results and discussion?

Line 232: insert space… tenderness [11,22].

Line 238: insert space… beef [23],

Line 238: [23]. In

Line 239: insert space…. 3.2 kg-3.9 kg

Line 249: remove indent space

Line 249: use italic text format for the subsection

Line 251: insert indent space

Line 258: CO2-eq

Line 260: insert space… Administration [7],

Line 262: insert space… 120 kg

Line 263: insert space… 16.55 kg

Line 263: CO2-eq

Line 263: insert space… 1 kg

Line 263: CO2

Line 264: insert space… 1 kg

Line 266,267,270,275: CO2

Line 268: insert space…. 26 months

Line 275: insert space… 1 kg

Line 276: insert a dot at the end of the table title

Line 276: in table content, insert a speace… 1 kg

Line 279: delete space

Line 280: remove indent space

Line 280: use italic text format for the subsection

Line 291: delete space… more.

Line 294,296: CO2-eq

Line 298: 13% ¿s? arise

Line 300: insert space… this [3,4].

Line 302: insert space… 1 kg

Line 308: remove indent space

Line 308: use italic text format for the subsection

Line 309: use LD and BF in table content, the information was previously abbreviated

Line 312: did you mean Conclusions?

Line 317: CO2-eq

Note: it is necessary to use the correct format for each of the references, you can consult the guide of authors (Microsoft word template). For example:

1.          Author 1, A.B.; Author 2, C.D. Title of the article. Abbreviated Journal Name Year, Volume, page range.

Author 1, A.; Author 2, B. Title of the chapter. In Book Title, 2nd ed.; Editor 1, A., Editor 2, B., Eds.; Publisher: Publisher Location, Country, 2007; Volume 3, pp. 154–196.

Author Response

Thank you for all your review comments.

I modified this manuscript following your comments.

And, detailed responses to some corrections were noted in the form of memos in the manuscript.

Reviewer 2 Report

Comments and Suggestions for Authors

The paper “Greenhouse gas emission and reduction of bovine toughness by localized electrical stimulation” focus on changes in meat quality produced by ageing and electrical stimulation as well as greenhouse gases emission during cattle breeding. The title might be modified to better specify the content of the paper. In meat science the term tenderness is used more frequently than toughness – please revise. The abstract section is not informative and should be rewritten. The main flaw of the work is the statistical analysis section. The results should be analysed using different statistical methods instead one-way analysis of variance (two or three-way ANOVA or variance components) to indicate the effect of electrical stimulation and ageing. I suggest comparing mean values obtained for different treatments such as control 2 days, ES 2 days, Control 14d, ES 14d from a particular muscle using the RIR Tuckey test instead of Duncan’s test to compare mean values. The section that describes the methodology of calculating the CO2 emission is not clear to me and needs to be described in a more clear way. The structure of the manuscript should be improved – there are some paragraphs in the Results section which are discussed and should be placed in the Discussion section. I don’t feel competent to evaluate the section about CO2 emission reduction. Below I enclosed detailed comments referring to the manuscript.

Abstract:

Line 9 „We examined the effects of partial electric stimulation on the quality of and carbon emissions..” please correct by adding the noun – the quality of what? Please rephrase the whole sentence to make it more clear, concise – divide it into separate sentences, 2 at least

The whole abstract should be rephrased with a clearly stated aim of the study, materials and methods and results. From the current form, it is unclear what the aim, the age of animals, how much it was shortened or if the ageing process was shorted (to how many days?). Please provide the most relevant results in the abstract.

Introduction

Line 34 “GHG” please provide the explanation for the abbreviation where used for the first time

Line 66 “Electrical” should be lower case letter “electrical”

Materials and methods

Line 81 it is advised not to use the name “longissimus dorsi”. Use “longissimus lumborum et thoracis” instead. To find more information please consult https://doi.org/10.1016/0309-1740(90)90010-4

Please correct the whole paper

Line 105-111 please provide more information about the number of repetitions, the surface used for colour determination – what do you mean by “each surface”?

Line 248 “4C” the degree symbol is missing

Line 147 what was a sample weight, how many samples were prepared?

Line 158 please provide the information about WBSF test parameters

Point 2.7. The assumption seems unclear to me, e.g. “Trade meat amount per 1kg of live cattle (kg) = (1 x B)/100” what is B? the one from Table 1? B. Live weigh (kg)? So, when we take 26 months, B=655.9 kg, trade meat amount is 6.5 kg/1kg of live cattle? What does it mean? Please explain. I don’t understand this part of the paper at all.

Statistical analyses – I recommend applying two-way ANOVA or variance components analysis – because there is more than one factor – in fact, there were three – ageing, electrical stimulation and two muscles. Therefore table 2 is not informative and should be changed for the one which covers the effect of muscle, ageing and electrical stimulation.

Table 2 – pH why the value for control  “5.53b” has only b letter while the value for ES “5.53ab” has ab?

Result section – please remove the discussion to the next section, e.g. lines 185-188, 188-191, 195-196, 230-246.

Through the paper “ageing” and “aging” are used. Please uniform.

Section 3.3 I do not really understand the information – the meat is stored in shop anyway and age during the time of distribution. Line 292 “When 1 kg of meat is aged for a single day under supermarket storage conditions, 0.0009 292 kW of electricity is used [19]” the meat is stored in a shop before it is sold anyway.

Line 323-325 “Localized electrical stimulation 323 was observed in this study to have significantly decreased the initial shear-force of BF 324 muscle that was low in intramuscular fat and to have accelerated beef aging by reducing 325 the aging period by more than 14 days.” The phrase is not clear. What do you mean by reducing the ageing period by more than 14 days? The meat quality was evaluated on days 2 and 14th, so is your conclusion that ageing is not necessary? In my opinion that conclusion is wrong because WBSF was lower on 14d than on 2nd day in longissimus lumborum at thoracis.

Although the CO2 emission reduction was calculated Authors discussed it in a way which might mislead the reader and make them think that the experiment was done using meat from animals of different ages – in the last section of the paper. Please make it clear it was only theoretical assumptions.

In the manuscript section “Conclusions” should be placed with a take-home message formulated based on the results of the study, i.e. the effect of local electrical stimulation on the quality of beef and usefulness of the process taking into account CO2 emission as well. I suggest deleting the part about CO2 emission with respect to the age of animals because no evidence of the meat quality was presented.

Author Response

Thank you for all your review comments.

I modified this manuscript following your comments.

And, detailed responses to some corrections were noted in attached file.

Reviewer 3 Report

Comments and Suggestions for Authors

The information in this manuscript is available from other literature sources, making the study unnecessary. A native English speaker should review the manuscript and correct the language and grammar errors. Some information on procedures to determine their appropriateness and to allow duplication of the experiments by other scientists was missing.

Line(s)      Comment

10              More details on the electrical stimulation, especially the voltage and when it was applied, is needed.

10-14        Reporting of the specific values obtained by electrical stimulation is desired.

15              The means of determining the “delicious” trait of the beef should be described.

15-16        Specific information on how much shorter breeding period and degree of lowering of finishing cattle is needed.

25-29        Reference(s) are needed to substantiate this information as factual.

31              The term “shedding technologies” is not generally familiar in its meaning.

41-43        Reference(s) needed for the Korean beef grading standards.

44-45        “increases, but there is a decrease in daily”

48              “from highly marbled beef.”

49-50        Reference(s) are needed to substantiate this information as factual.

60-61        Reference(s) needed to validate that consumers perceive a shear-force of 3.2 kg or less as tender.

67              The term “flue improvement technologies” is not understood.

73-74        The difference in using electrical stimulation in this study and previous studies using electrical stimulation should be described here.

87              The temperature of the chilling room and the rate of temperature decline in the beef carcasses should be given.

91              More details on the division procedure: the material, vapor permeability or thickness, and source of the packaging materials; and the level of vacuum, vacuum packaging equipment model, and manufacturer of the packaging equipment must be given.

94              “Aging acceleration though electrical stimulation”

110           “color meter”

112           The aperture opening size, optical geometry, observer angle should also be given.

122           It should be documented that 5 hours is sufficient for drying samples.

124           It should be documented that 6 hours is sufficient to extract all of the fat from the prepared samples.

125           The method of evaporating the petroleum ether should be described.

138           “homogenate was filled to 30 mL”

144           “Absorbance”

150           The vapor permeability or thickness and the source of the polyethylene bag should be given.

153           The method of water removal from the cooled sample should be described.

155           The method of collecting the 1.25 cm diameter cores should be described.

160-168   This section is very confusing and must be rewritten for clarity. The use of 12 to 18 year-old references as being relevant to today must be justified.

170           If the carcasses and two muscles were randomized to electrical stimulation (ES and no-ES) and two aging periods (2 and 14 d) and as indicated by the letters indicating differences in Table 2, then a two-way analysis of variance with any interaction effect should have been used. If the design was a randomized block, then the experimental design and statistical model should be described.

189-190   Reference(s) are needed to support the statements that BF muscle has a higher density of muscle bundles than LD muscle and the structure was weakened by electrical stimulation since the density and structural integrity were not measured in the present study.

196           “food storage stability.”

Table 2     “Crude fat”

232-236   Reference(s) are needed to substantiate that this information is factual.

251-254   It must be substantiated that the types of Hanwoo cattle produced now are not different in genetics, performance, and meat quality than those that were produced in 2005 for this reference to be valid.

255           “breeding” generally refers to the time for an animal to reach sexual maturity and/or the amount of time for the dam to become pregnant, not the “feeding” time for calves to reach the desired slaughter weight.

256-258   It must be substantiated that Hanwoo feeding systems are similar to those for Japanese Wagyu beef for this calculation to be used in the current study.

259-262   It must be substantiated that the increase of liveweight in 5 months now is the same as the liveweight increase in 5 months that occurred in 2005.

268-271   This information could be gained from the scientific literature without having to conduct this research project.

271-272   The 1 kg decrease in WBSF was achieved only with ES for the Biceps femoris at 2 days of aging and not for BF at 14 days or for Longissimus dorsi.

282-285   Reference(s) needed to substantiate this information as factual since flavor components and the actual amount of energy consumption were not measured in this study.

293-294   A reference for the Korean Ministry of Environment data must be given.

298-299   This dependent clause is incorrect as livestock products are heated, dried, fermented, and/or preserved by other means and sold as shelf-stable items.

Table 5     The unit of measurement for the effects of acceleration of aging must be given.

313-332   This section has an incomplete discussion of the results and combines conclusions that are not necessarily indicated by the results shown in the tables.

345-404    There are inconsistencies in the references, with incomplete citation information on some references, inconsistent formats of author and journal title capitalizations, differences in journal title abbreviations, and inconsistent punctuation in the references.

Comments on the Quality of English Language

There are some difficulties that should be corrected by a native English speaker.

Author Response

Thank you for all your review comments.

I modified this manuscript following your comments.

And detailed responses to some corrections were noted in the attached file.

Thank you again. 

Round 2

Reviewer 1 Report

Comments and Suggestions for Authors

Dear authors, in a second review, it is determined that it is necessary to attend to the following recommendations:

Note 1: throughout the document the temperature units and others are abbreviated, however, what is the reason why the time units are not presented in an abbreviated form?

Section 2.3.: What were the methods used to determine pH and color coordinates?

Line 144: 120 mL

Line 256-262; 281-284: information should not be discussed in the results section

Note 2: In the list of references, the correct format was not used to cite most of these. Review the Microsoft word template: https://www.mdpi.com/files/word-templates/foods-template.dot

Reviewer 2 Report

Comments and Suggestions for Authors

Dear Editor and Authors,

The manuscript was significantly modified and the Authors responded to all comments raised in the review. However, it is a pity that the Authors did not follow my guidelines to indicate the effect of electrical stimulation and ageing separately. Nevertheless, still there are some issues to be improved or explained. Below I attached details.

Lines 11-12 „carbon dioxide (CO2) reduction achieved through a shortened fattening period using the localized electrical stimulation”  - the expression ”shortened fattening period using the localized electrical stimulation” is misleading – you just shortened fattening period and used electrical stimulation on meat post slaughter not as suggested by the sentence shortened fattening period using ES. Please rewrite.

Line 12-14 “As a results, the application of localized electrical stimulation treatment did not affect in significant differences in the impact on the chemical stability of both LT and BF muscles” In my opinion, the sentence should be simplified e.g.  The application of localized electrical stimulation treatment did not significantly affect the chemical stability of both LT and BF muscles. Please also specify what you mean by “chemical stability” because it is unclear.

Line 19-21 “The result of estimating CO2 mitigation from shorter feeding period on Hanwoo steer, longer fattening period increased intramuscular fat and decreased shear force, but it also led to higher CO2 emissions” In my opinion the sentence is too complicated and misleading because it states that the estimation enables to predict fat content and shear force, which is not true. and should be shortened to e.g. “The estimation of CO2 production showed that longer fattening period leads to higher CO2 emissions”. 

Lines 21-23 Please add a value of CO2 reduction e.g. in %

Lines 24-25 “in the case of BF muscle, the maturation period was shortened by more than 14 days” In my opinion the conclusion is somehow misleading – you had only 2 ageing times 2 and 14 days therefore saying that “the maturation period was shortened by more than 14 days” is not correct. Please rewrite.

Line 28 “achieved through electrical stimulation and a shortened fattening period” Please keep the chronology – the first was shortened fattening period and then ES was applied to carcasses. Please rewrite.

Eq. 1. “Trimmed meat amount per 1 kg of a live cattle (kg) = (1 x B)/100” This makes no sense because if B is 655.9 kg it gives 6.559 kg per 1 kg of live cattle which is a non-sense. Using the result in further calculations gives wrong results

Eq. 2. “Amount of carbon dioxide emitted per 1 kg of the trimmed beef” How trimmed meat can emit CO2? Need to be explained or rewritten

Table 1 The column 31-26 months – E value should be -0.98 

Table 2

Use * next to L*, a*, b*

Line 450-451 “In other words, the electrical stimulation treatment showed the effect of shortening the ageing period of BF muscle (muscle with low intramuscular fat contents) by more than 14 days.” As mentioned earlier the sentence is misleading, please delete it.

Reviewer 3 Report

Comments and Suggestions for Authors

Excessively wordy in many sentences. Sources of information provided as fact must be documented with reference(s) or detailed explanations. A native English speaker should review the manuscript and correct the language and grammar errors. The missing information on procedures prevent determining if they are appropriate and do not allow duplication of the experiments by other scientists was missing.

Line(s)      Comment

13-14        “localized electrical stimulation treatment (45 V) did not affect chemical stability”

19-21        “The longer fattening period increased intramuscular fat and decreased 20 shear force, but it also led to higher CO2 emissions.”

87-90        Poorly worded.

90-93        Reference(s) needed to substantiate that this information is factual.

94-97        Reference(s) needed to substantiate that this information is factual.

111-115   In a well designed experiment, the ES and control treatments would have been randomly assigned to left and right sides.

122           The material and vapor permeability or thickness of the packaging, level of vacuum, and model and manufacturer of the vacuum packager must be given.

142-144   The aperture opening size, optical geometry, observer angle must also be given.

182           Incomplete sentence.

182-184   The term “meat stake” is not understood.

185           The vapor permeability or thickness of the polyethylene bag must be given

217           The different font and word spacing make it impossible to determine the meaning of the sentence.

226-227   Reference(s) needed to substantiate that the pH of this meat remained within a normal range for beef.

232-235   This repetitive wording should be deleted.

247           Grammar error.

365-367   Grammar error.

373-375   Incomplete sentence.

391-393   Reference(s) or an explanation to provide the basis for justifying this statement is needed.

Comments on the Quality of English Language

Language errors were present in the revised wording in the manuscript.